# An Efficient Private GPT Never Autoregressively Decodes

**Zhengyi Li** [1 2] **Yue Guan** [1] **Kang Yang** [3] **Yu Feng** [1] **Ning Liu** [1] **Yu Yu** [1 2] **Jingwen Leng** [1 2] **Minyi Guo** [1 2]

## Abstract

The wide deployment of the generative pre-trained transformer (GPT) has raised privacy concerns for both clients and servers. While cryptographic primitives can be employed for secure GPT inference to protect the privacy of both parties, they introduce considerable performance overhead. To accelerate secure inference, this study proposes a public decoding and secure verification approach that utilizes public GPT models, motivated by the observation that securely decoding one and multiple tokens takes a similar latency. The client uses the public model to generate a set of tokens, which are then securely verified by the private model for acceptance. The efficiency of our approach depends on the acceptance ratio of tokens proposed by the public model, which we improve from two aspects: (1) a private sampling protocol optimized for cryptographic primitives and (2) model alignment using knowledge distillation. Our approach improves the efficiency of secure decoding while maintaining the same level of privacy and generation quality as standard secure decoding. Experiments demonstrate a $2.1\times \sim 6.0\times$ speedup compared to standard decoding across three pairs of public-private models and different network conditions.

## 1. Introduction

The generative pre-trained transformer (GPT) (Vaswani et al., 2017) has brought significant advancements in various machine learning tasks, such as coding (GitHub, 2025) and chatbots (Leiter et al., 2023). To deploy a GPT application, either the client needs to upload private data or the model owner needs to send its private model to the client. As GPT increasingly handles sensitive data and tasks, privacy be-

comes a major concern. Recent works (Hao et al., 2022; Lu et al., 2025; Hou et al., 2023; Pang et al., 2024)leverage secure two-party computation (2PC) to enable privacy-preserving GPT inference. These studies allow the client and server to jointly execute inference on encrypted inputs and models utilizing cryptographic techniques such as homomorphic encryption (HE) (Gentry, 2009) and multi-party computation (MPC) (Yao, 1986). Consequently, the client only learns the inference results without access to model weights, and the server knows nothing.

However, privacy protection incurs significant computational and communication costs for both the linear and nonlinear layers. Linear layers involving matrix multiplication are usually computed through HE, which requires substantial computation. Complex nonlinear layers, such as GELU and softmax, require numerous rounds of communication and data transmission. To accelerate secure GPT inference, prior works focus on optimizing cryptographic protocols (Hao et al., 2022; Lu et al., 2025; Pang et al., 2024; Kim et al., 2024) or modifying GPT architectures to tailor the cryptographic primitives (Li et al., 2022; Zeng et al., 2023; Zimerman et al., 2024; Ran et al., 2023; Li et al., 2024a). Despite these efforts, achieving efficient secure inference remains a major challenge.

This study explores an orthogonal approach by utilizing public GPTs to accelerate secure GPT inference, mainly during the decoding phase. We notice the availability of powerful public GPT models within the GPT community (Wolf et al., 2019). These models share some linguistic and logical knowledge with private models (Chen & Gao, 2022; Zhao et al., 2023). Since this shared knowledge is inherently non-sensitive and publicly disclosed, a more efficient approach is to only securely compute the exclusive parts of the private model instead of all. However, few works have explored this direction due to the lack of interpretability of the GPT models (Friedman et al., 2024; Wen et al., 2024), making it challenging to separate shared knowledge.

This work introduces an observation of the performance characteristics of secure decoding to utilize public models for acceleration without comprising privacy. Standard secure decoding forwards only one token per decoding step. We observe that securely forwarding single and multiple tokens takes similar latency. This motivates us to treat the

---

[1]Shanghai Jiao Tong University [2]Shanghai Qizhi Institute [3]State Key Laboratory of Cryptology. Correspondence to: Jingwen Leng <leng-jw@sjtu.edu.cn>, Kang Yang <yangk@sklc.org>, Yu Yu <yyuu@sjtu.edu.cn>.

*Proceedings of the 42$^{nd}$ International Conference on Machine Learning*, Vancouver, Canada. PMLR 267, 2025. Copyright 2025 by the author(s).

model as a whole to avoid the difficulties arising from distinguishing shared knowledge. Inspired by the speculative decoding (Leviathan et al., 2023; Miao et al., 2024b), we propose a Public decOding and Secure verificaTion (POST) approach. Specifically, the client employs the public model to generate a set of draft tokens, which are then fed into the private model to verify their acceptance in a single decoding step. Given the shared knowledge between public and private models, it is likely that both models produce identical results when generating simple tokens or common phrases. This allows multiple tokens to be accepted in a decoding step, reducing the required steps and the amortized cost per token.

In POST, the reduction in decoding steps depends on the acceptance ratio of tokens proposed by the public models, which we further improve through two aspects. (1) Secure speculative sampling for soft matching: strict matching between tokens generated by the public and private models causes unnecessary rejections, such as semantically equivalent tokens that are expressed differently. Therefore, we adopt the speculative sampling algorithm (Leviathan et al., 2023; Cai et al., 2023), which applies "soft" matching to increase the acceptance ratio while ensuring the generated tokens follow the private model's output distribution. Since the computation in the speculative sampling algorithm is unfriendly to the cryptograph, we propose a protocol to optimize the cryptographic-unfriendly operations to achieve negligible overhead. (2) Knowledge distillation for model alignment: discrepancies between the public and private models reduce the acceptance ratio of draft tokens. To mitigate this, we align the two models through knowledge distillation, further increasing the acceptance ratio.

In summary, this paper makes the following contributions:

- We present a novel observation: the latency of secure GPT decoding is insensitive to input length. Based on this, we propose the POST approach to integrate public GPT models into secure inference for acceleration while maintaining privacy and accuracy. This approach broadly applies across different cryptographic protocols and GPT models, where we observe similar insensitivity.

- We further enhance the efficiency of POST through two aspects: an optimized protocol that securely samples tokens from two models and the alignment of models using knowledge distillation.

- We evaluate the performance under two network conditions and three pairs of private and public models, including Vicuna-7B and LLaMA-68M&160M, FLAN-T5-XL and T5-efficient-small&base, and FLAN-T5-XL and FLAN-T5-small&base. Our approach shows $2.1\times \sim 6.0\times$ speedup than the standard secure decoding.

## 2. Background

This work aims to accelerate the secure decoding of generative pre-trained transformers (GPT). We present the necessary background on secure GPT inference and standard GPT decoding. Additional background and related works are in the Appendix 6.

### 2.1. Secure Two-Party Inference

Under the semi-honest threat model, where the corrupted party adheres to the protocol but may attempt to extract more information than permitted, secure two-party GPT inference guarantees the client receives inference results without accessing the model weights and the server remains oblivious to the client's private input (Juvekar et al., 2018; Hao et al., 2022; Lu et al., 2025; Pang et al., 2024; Li et al., 2024b). Existing approaches typically employ hybrid protocols, combining homomorphic encryption (HE) and multi-party computation (MPC) based on the nature of each operation. HE is commonly used for linear operations, such as multiplication, which incurs intensive computation but requires minimal communication. MPC is generally employed for nonlinear operations, such as comparison. MPC involves multiple rounds of communication and extensive transmitted data. Below, we provide a brief background on cryptographic primitives relevant to this work.

**Additive Secret Sharing.** Parties usually hold activations and model weights through secret sharing. In a two-party setting, additive secret sharing over the ring $\mathbb{Z}_{2^\ell}$ is defined as follows: for a given value $x \in \mathbb{Z}_{2^\ell}$, two random shares, $\langle x \rangle_c \in \mathbb{Z}_{2^\ell}$ and $\langle x \rangle_s \in \mathbb{Z}_{2^\ell}$, are uniformly sampled such that $x = \langle x \rangle_c + \langle x \rangle_s \mod 2^\ell$ holds. Here, $\langle x \rangle_c$ and $\langle x \rangle_s$ are held by the client and the server, respectively.

**Homomorphic Encryption.** The lattice-based additive HE scheme (Rathee et al., 2019) is proven secure under the ring learning with errors (RLWE) assumption (Lyubashevsky et al., 2010). HE enables one party to perform computations on another party's encrypted data without necessitating access to the decryption key. HE typically employs Single Instruction Multiple Data (SIMD) techniques to reduce amortized overhead (Brakerski et al., 2014; Juvekar et al., 2018). Multiple plaintext values are encoded into one polynomial ring, enabling homomorphic operations to be applied to all encoded values in parallel.

**Oblivious Transfer.** Oblivious Transfer (OT) is the building block for various nonlinear operations, such as comparison and truncation (Rathee et al., 2020; Ma et al., 2023). Let $\binom{k}{1} - \mathrm{OT}_\ell$ denote the 1-out-of-$k$ OT functionality, which is a generalization of the 1-out-of-2 OT. The sender's inputs consist of $k$ strings, $m_0, \ldots, m_{k-1}$, each of length $\ell$ bits, while the receiver's input is an index $i \in [k]$. The receiver obtains $m_i$ from the functionality, and the sender

receives nothing. The $\binom{k}{1} - \mathrm{OT}_\ell$ protocol requires two rounds of interaction and involves a total communication costs of $k\ell + \log_2 k$ bits (Yang et al., 2020).

## 2.2. Prefill and Decoding of the GPT

The standard GPT generation process consists of two main phases: the prefill phase and the decoding phase. The prefill phase processes all tokens from the client's input prompt in a single forward pass, producing the output distribution for the first generated token. The decoding phase generates one token per decoding step in an auto-regressive manner. For a prefix $x_{<t}$ with $t - 1$ tokens, which include both the input prompt and previously generated tokens, the forwarding of $x_{<t}$ generates the output distribution $p(x \mid x_{<t})$ of the $t_{th}$ token and samples from it. When $t$ is clear or unimportant, $p(x)$ is used simply for representation. To avoid repeatedly forwarding all previous tokens when generating a new token, the KV cache (Dettmers et al., 2022) stores the keys and values associated with past tokens, such that each decoding step only forwards one token.

Since secure decoding accounts for the majority of computation time (Liang et al., 2024), this work focuses on improving its efficiency. A detailed explanation of the prefill and decoding phases is provided in Appendix A.

## 3. Motivation

The public model shares partial knowledge with the private model. To utilize such knowledge for acceleration without comprising privacy, we introduce an observation on the performance characteristics of securely computed GPT layers.

### 3.1. Shared Knowledge of Public and Private GPTs

**Publicly Available GPTs.** The availability of numerous public pre-trained models (Wolf et al., 2019; Touvron et al., 2023; Chung et al., 2024) introduces a distinctive characteristic compared to other privacy-related scenarios: the knowledge embedded in the private GPTs is not entirely private. The linguistic knowledge (grammar, syntax, and common facts) and logical abilities (understanding and reasoning) can also be captured by the public model and are partly shared between public and private models (Chen & Gao, 2022; Zhao et al., 2023). As supporting evidence, the public models also present satisfying performance on various tasks (Analysis, 2025), although less powerful than private models.

Given that public GPTs have revealed shared knowledge, an ideal secure inference for the private GPT should allocate costly cryptographic computation only to exclusive knowledge while employing inexpensive plaintext computations for shared knowledge. However, due to the lack of inter-

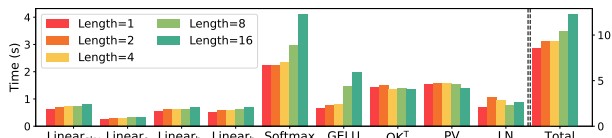

*Figure 1.* Latencies against different input lengths. The bandwidth and one-way delay are 1000 Mbps and 10 ms.

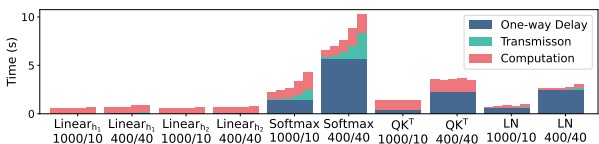

*Figure 2.* The latency breakdown of some layers. The second row of the x-axis ticks represents bandwidth and one-way delay. The bars corresponding to the same x-axis ticks illustrate input lengths 1, 2, 4, 8, and 16.

pretability in model weights and activations (Friedman et al., 2024; Wen et al., 2024), clearly separating shared knowledge between public and private models is challenging, and any reckless use of plaintext computations may introduce unforeseen risks. Therefore, current secure inference methods require that all computations remain private.

### 3.2. Insensitivity of Latency to the Input Length

To utilize the public GPT model for acceleration, we introduce a unique performance characteristic observed in two-party secure decoding, which allows us circumvent the challenge of separating the knowledge between the public and private models. Figure 1 illustrates the decoding latencies for different input lengths using the open-source model FLAN-T5-XL (Chung et al., 2024). The latencies are profiled on the well-established framework SecretFlow-SPU (Ma et al., 2023). In standard decoding, GPT generates only one token per step. For such minimal input length one, we find that gradually increasing the input length has almost no impact on latencies. The latencies remain similar when the input length increases from 1 to 4. As the input length continues to increase to 8 or 16, latency only experiences a sub-linear increase. Consequently, the total latencies of a Transformer decoder on input lengths 8 and 16 are only $1.2\times$ and $1.5\times$ greater than that of input length of 1. In Appendix E.1, we observe similar insensitivity to input length across various network conditions, GPT sizes, and underlying cryptographic protocols.

To understand why such insensitivity generally holds, we decompose the latency of each layer into the one-way delay, computation time, and transmission time, as illustrated in Figure 2. We analyze how each component contributes to

the insensitivity of the overall latency.

**One-way Delay.** The one-way delay is a fixed latency per communication round due to propagation time and hardware processing. Time spent on the one-way delay remains constant, as the number of communication rounds is independent of the input length. Secure evaluation generally necessitates multiple rounds of communication, especially nonlinear protocols that require more than a hundred rounds. The time spent on one-way delays constitutes a substantial portion of the overall latency, directly contributing to the overall insensitivity to variations in input length.

**Computation Time.** The computation time demonstrates a small variation in input length. The computation time is mainly spent on homomorphic operations, including matrix multiplication in the linear layer and Hadamard multiplication of the approximated polynomials in the nonlinear layer. The insensitivity of computation time can be attributed to two aspects. First, larger input lengths typically enable better parallelization and more efficient hardware utilization. Second, increasing the input length improves the efficiency of SIMD operation in HE (Hao et al., 2022; Lu et al., 2025). To ensure security, SIMD operation typically encodes 8192 activation values into the ciphertext polynomial. However, with an input length of 1, the number of activation values is often smaller than 8192. For instance, the hidden dimension of FLAN-T5-XL is 2048, and smaller models like GPT-2 have dimensions as low as 768. Larger input lengths can better saturate all slots in the polynomial ring to enhance computational efficiency. As Figure 2 shows, homomorphic operations constitute a significant portion of overall latency and contribute to the overall insensitivity.

**Transmission Time.** The relationship between transmission time and input length differs between HE and MPC. For HE, the transmission size increases sub-linearly due to the efficiency of SIMD. As input length increases, multiple embedding vectors are packed into a single ciphertext, which does not change the overall transmission size. In contrast, the transmission size associated with MPC generally increases linearly with input length. Overall, compared to one-way delay and computation time, transmission time is more sensitive to input length variations. However, transmission time is not the primary bottleneck for extremely small input lengths, and its impact on the overall latency increase remains limited.

# 4. Public Decoding and Secure Verification

Based on the observation in Section 3, we first propose a new approach for secure GPT decoding. Then, we further improve its efficiency from two aspects.

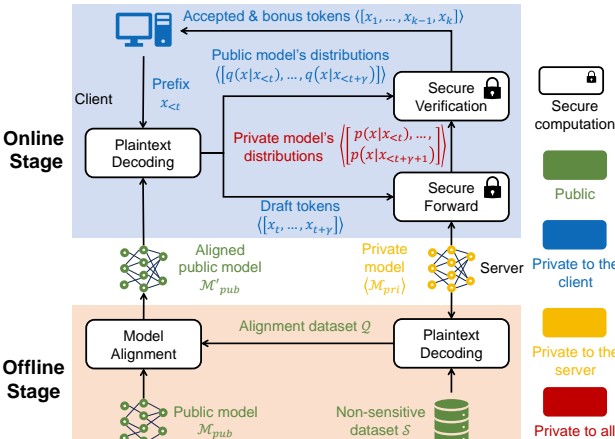

*Figure 3.* The overview of the public decoding and secure verification approach. $\langle \cdot \rangle$ indicates the data are encrypted during computing, such as using the secret sharing, and is only visible to the data owner.

## 4.1. Overview

We propose an approach to leverage knowledge from the public model to enhance efficiency: Public decOding and Secure verificaTion (POST), as illustrated in Figure 3. POST includes online and an one-time offline stages.

In the online stage, the client holds a public model $\mathcal{M}'_{pub}$, while the server holds a private model $\mathcal{M}_{pri}$. The client also holds the prefix $x_{<t}$, consisting of $t - 1$ tokens, which include the input prompt and previously generated tokens. In prior secure inference works, only the next token is generated in a single decoding step. This results in only one vector being forwarded in the secure forward, leading to inefficiencies demonstrated in Section 3.2. In contrast, POST first lets the client autoregressively sample $\gamma$ draft tokens from the public model's output distributions $x_1 \sim q(x \mid x_{<t}), \cdots, x_\gamma \sim q(x \mid x_{<t+\gamma-1})$. Subsequently, both parties securely forward all draft tokens in a decoding step to generate private model's output distributions $\langle p(x \mid x_{<t}) \rangle, \cdots, \langle p(x \mid x_{<t+\gamma}) \rangle$, which are secret shares on the ring $\mathbb{Z}_{2^\ell}$. Then the **secure verification** (Section 4.2) adopts the speculative sampling algorithm (Leviathan et al., 2023) to determine which tokens can be accepted based on both the public and private models' output distributions. We give specialized optimizations to securely execute speculative sampling, minimizing its overhead. Tokens before the first rejection are kept, while those after are discarded. Since the private model accepts all tokens up to the first rejection, the first rejected token can be re-sampled from the private model's adjusted output distribution to obtain an additional "bonus" token.

In the offline stage, we also perform **model alignment** (Section 4.3) to align the public model's output distribution with

that of the private model through knowledge distillation. This can further increase the acceptance ratio of draft tokens and enhance performance improvements.

Our approach offers following advantages.

- Consistent Speedup: In worst case where all draft tokens are rejected, POST's secure forward still generates a bonus token, the same as the previous secure decoding methods. In more general cases, easily predictable draft tokens are likely to be accepted by the private model. Therefore, forwarding multiple tokens incurs a cost comparable to forwarding just one token, while allowing for the generation of at least one token in a single step, thus reducing the amortized cost per token.

- Security Guarantees: POST ensures that the client obtains only the information permitted in standard secure inference protocols, while the server remains oblivious to the client's private input data.

- Accuracy Preservation: The token generated in POST precisely maintains the output distribution of the private model in standard secure inference, ensuring no accuracy degradation.

- Practical Implementation: POST imposes no change or fine-tuning on the private models and can be seamlessly integrated with existing secure inference methods.

- Future Scalability: The continuous advancement of public model capabilities presents significant potential of the acceptance ratio of the draft tokens, ensuring sustained performance improvement as the acceptance ratio increases.

The following sections detail our designs for secure verification and model alignment to enhance efficiency.

### 4.2. Secure Verification

**Naive Sampling from Two Distributions.** To determine whether draft tokens are accepted, a naive way is to sample from the output distribution of private models and only accept matched tokens. However, this method results in a low acceptance ratio. In many cases, tokens with similar meanings may exhibit similar output distribution densities, yet these proposals may still be rejected due to the "hard" matching.

**Speculative Sampling.** Therefore, we employ a "soft" matching called speculative sampling (Leviathan et al., 2023; Cai et al., 2023). Speculative sampling ensures that the tokens sampled from $p(x)$ and $q(x)$ are distributed identically to those sampled from $p(x)$ alone. Specifically, given $\gamma$ draft tokens $[x_1, \cdots, x_\gamma]$ sampled from the distributions

---

**Algorithm 1** Privately Reject Draft Tokens

**Input:** $P_c$ holds public model's distributions $q(x \mid x_{<t}), \cdots, q(x \mid x_{<t+\gamma-1})$ and samples $\gamma$ draft tokens $x_1, \cdots, x_\gamma$. $P_s$ and $P_c$ hold the secret sharing $\langle p(x \mid x_{<t}) \rangle, \cdots, \langle p(x \mid x_{<t+\gamma}) \rangle \in \mathbb{Z}_{2^\ell}$ of the distributions generated by the private model. The vocabulary size is $\mathcal{V}$. The $[\gamma]$ means $[0, 1, \ldots, \gamma]$.

1: $P_c$ sets $\mathbf{Q} = \begin{bmatrix} q(x \mid x_{<t}) \\ \cdots \\ q(x \mid x_{<t+\gamma-1}) \end{bmatrix}$. $P_s$ and $P_c$ set $\langle \mathbf{P} \rangle = \begin{bmatrix} \langle p(x \mid x_{<t}) \rangle \\ \cdots \\ \langle p(x \mid x_{<t+\gamma}) \rangle \end{bmatrix}$.

2: $P_c$ generates a random vector $\mathbf{R}_{mul} \sim U(0,1)^{\gamma \times \mathcal{V}}$.

3: $P_s$ sets his share of $\langle \mathbf{S} \rangle_s = -\langle \mathbf{P} \rangle_s$ and $P_c$ sets her share $\langle \mathbf{S} \rangle_c = \mathbf{Q} \cdot \mathbf{R}_{mul} - \langle \mathbf{P} \rangle_c \mod 2^\ell$.

4: **for** $i \in [\gamma]$, in parallel **do**

5:     $P_s$ generates a random vector $\mathbf{r} \sim \mathbb{Z}_{2^\ell}^\gamma$, and masks his share as $\langle \hat{\mathbf{S}}[i,:] \rangle_s = \langle \mathbf{S}[i,:] \rangle_s - \mathbf{r}[i] \mod 2^\ell$.

6:     $P_c$ chooses the secret share by $\langle \hat{\mathbf{S}}[i, x_i] \rangle_s \sim \binom{\mathcal{V}}{1} - \mathrm{OT}_\ell(\langle \hat{\mathbf{S}}[i,:] \rangle_s, x_i)$.

7:     $P_s$ sets his share $\langle \mathbf{s} \rangle_s[i] = \mathbf{r}[i]$ and $P_c$ sets her share $\langle \mathbf{s} \rangle_c[i] = \langle \mathbf{S}[i, x_i] \rangle_c + \langle \hat{\mathbf{S}}[i, x_i] \rangle_s \mod 2^\ell$.

8: **end for**

9: $P_s$ and $P_c$ compute shares of boolean vector $\langle \mathbf{n} \rangle = \mathcal{F}_{less}(0, \langle \mathbf{s} \rangle)$ and open $\mathbf{n}$ to $P_c$.

10: $P_c$ appends an element one at the end by $\mathbf{n} = \mathbf{n} + [1]$

11: $P_c$ computes $k = \min(\{i \mid i \in [\gamma], \mathbf{n}[i] = 1\})$.

12: $P_c$ chooses $\langle p_k(x) \rangle_s \sim \binom{\gamma+1}{1} - \mathrm{OT}_\ell([\langle p(x \mid x_{<t}) \rangle_s, \cdots, \langle p(x \mid x_{<t+\gamma}) \rangle_s], k)$.

**Output:** $P_c$ obtains index $k$ and corresponding $p_k(x)$.

---

of the public model, each token $x_i$ is rejected according to the probability defined by:

$$\max\left(0, 1 - \frac{p(x_i \mid x_{<t+i})}{q(x_i \mid x_{<t+i})}\right). \tag{1}$$

The $p(x_i \mid x_{<t+i})$ represent the probability density of the $x_i$. If the first rejected token has $i < \gamma$, it is resampled from an adjusted distribution $p'(x) = \max(0, p(x \mid x_{<t+i}) - q(x \mid x_{<t+i}))$. If all draft tokens are accepted, the bonus token is directly sampled from the distribution $p(x \mid x_{<t+\gamma})$.

**Challenge of Secure Speculative Sampling.** The performance challenge lies in rejecting draft tokens according to the probability in Equation (1). The standard approach is to privately compute Equation (1) and then compare it with a random number within $[0, 1]$. However, the involved division and comparison are not friendly to the cryptographic primitives. Furthermore, the dimension of the output distributions is the vocabulary size $\mathcal{V}$ (typically $\mathcal{V} > 30000$), making probabilistic rejection in secret considerably slow.

To mitigate this issue, the proposed protocol is in Algorithm 1. Next, we explain two key designs in our protocol.

**Refactor the division into multiplication (lines 2-3).** To avoid the division $\frac{p(x_i)}{q(x_i)}$, we let the client generate a random matrix $\mathbf{R}_{mul}$ and execute Hadamard multiplication with $\mathbf{Q}$ locally. After the client subtracts $\langle \mathbf{P} \rangle_c$ from $\mathbf{Q} \cdot \mathbf{R}_{mul}$, both parties hold shares of the scores $\langle \mathbf{S} \rangle = \langle \mathbf{Q} \cdot \mathbf{R}_{mul} \rangle - \mathbf{P}$ mod $2^\ell$. For all draft tokens, the elements corresponding to the selected index $x_i$ are compared with zeros to generate boolean rejection decision $\mathbf{1}\{\mathbf{S}[i, x_i] > 0\}$ for $i \in [\gamma]$.

**Selection then Comparison (lines 4-9).** For elements $\mathbf{S}[i, x_i]$ that correspond to draft tokens, directly selecting their most significant bit (MSB) out for sign bit is not feasible. This is because the $\mathbf{S}$ are represented as secret sharing on the ring $\mathbb{Z}_\ell$. Thus, the parties must securely compute the carry bits from the lower $\ell - 1$ bits to obtain the correct MSB. This computation necessitates a $\binom{2^\ell}{1} - \mathrm{OT}_2$ (Rathee et al., 2020)[1]. Since the server cannot know which element is selected, both parties must securely compute the carry bits for all elements; then, the client selects the desired MSB through another $\binom{\mathcal{V}}{1} - \mathrm{OT}_1$. For each draft token, the total communication complexity is $\mathcal{O}(2 * \mathcal{V} * 2^\ell + \mathcal{V} * \ell + \mathcal{V} + \log_2 \mathcal{V})$. When selecting data, the communication complexity is linear with respect to the bit width. In contrast, when computing the carry bit, the bit width comes as the exponential complexity. Computing the carry bit of all elements incurs considerable unnecessary overhead.

To eliminate this unnecessary overhead, we first reconstruct shares of the interested elements and then compares them with 0. For each row of the score matrix $\mathbf{S}[i, :]$, the server masks his share by a common random value as $\langle \hat{\mathbf{S}}[i, :] \rangle_s = \langle \mathbf{S}[i, :] \rangle_s - \mathbf{r}[i]$. The client employs the index of the draft token to retrieve the share of the selected element through $\binom{\mathcal{V}}{1} - \mathrm{OT}_\ell$. The server remains unaware of which score share is retrieved yet retains the shares of the selected score, with the server holding $r$ and the client holding $\langle \mathbf{s} \rangle_c = \mathbf{S}[i, x_i] - r \mod 2^\ell$. Subsequently, the two parties evaluate comparisons on the selected element to derive rejection decision. In this manner, the overall communication complexity is $\mathcal{O}(\mathcal{V} * \ell + \log_2 \mathcal{V} + 2 * 2^\ell + \ell)$.

**Security Analysis.** In our private sampling, the server remains unaware of any client information. The only disclosed information consists of the rejection boolean values and the distributions used for re-sampling the first rejected token. As

---

[1]Existing works usually trades more communication rounds with reduced transmitted size by breaking one $\binom{2^\ell}{1} - \mathrm{OT}_2$ into $q$ instances of both secure AND operation and $\binom{2^m}{1} - \mathrm{OT}_2$, where $q * m = \ell$ (Rathee et al., 2020; Ma et al., 2023). We use a simplified comparison protocol for illustration, and this does not affect the conclusion.

the final output, the output distribution is supposed to be disclosed to the client for sampling tokens. Rejecting boolean values is a natural result once the client knows which token has been generated. In this way, the information revealed is no more than the standard decoding. All other information is in the form of secret shares and processed through cryptographic primitives. The security guarantee stems from the composability of cryptographic protocols. We put a more detailed analysis in Appendix D.

### 4.3. Aligning Public and Private Model

It is observed that, although fundamental linguistic knowledge is expected to be common across models, variations in training datasets and methodologies lead to biased output distributions between models. To improve the acceptance ratio, we propose aligning the public model with the private model through knowledge distillation (Li et al., 2022; Agarwal et al., 2024).

To perform the alignment, the client can choose a dataset independent of his privacy, such as a public corpus or anonymized data (Tang et al., 2024). This ensures that the queries can be accessed by the service provider and processed through plaintext inference. The client begins by querying the model using the standard decoding method and collects a dataset that includes the prompt and all output distributions, denoted as $\mathcal{Q} = \left\{ \mathbf{x}^{(i)} : \{ p(y_t^{(i)} \mid \mathbf{x}^{(i)}, y_{<t}^{(i)}) \}_{t=1}^{n_i} \right\}$, where $n_i$ is the response length for the $i_{th}$ prompt. Note that the general GPT API only returns the output distributions' topK (usually top 5) elements (Leiter et al., 2023). The client can only use these topK elements for alignment. On an input prompt $\mathbf{x}^{(i)}$, the client aligns the public and private models using the loss function

$$\ell(\mathbf{x}^{(i)}) = \sum_{t=1}^{n_i} D\left(p(y_t^{(i)}|\mathbf{x}^{(i)}, \mathbf{y}_{<t}^{(i)}) \| q(y_t^{(i)}|\mathbf{x}^{(i)}, \mathbf{y}_{<t}^{(i)})\right),$$
(2)

where $D(p\|q) = -\sum p \log q$ is the cross entropy. By minimizing this loss function, the public model learns to generate tokens that better align with the private model's output distribution, improving the acceptance ratio of the draft tokens.

**Security Analysis** The security of the model alignment is trivial to show since all used data are allowed to disclose. The query data used in the alignment are public corpus or anonymized data. These data are safe to be obtained by the server. The knowledge distillation only uses the topK elements of the output distribution of the private model, which are allowed to be revealed to the client.

**Server Provided Public Model.** Although the discussion lets the client select and align the public model, a more effective approach is to let the server provide an aligned public

*Table 1.* The acceptance ratio of tokens proposed by public models before and after the alignment (AL).

| Private Model | Public Model | Task | | | | | | | |
|---|---|---|---|---|---|---|---|---|---|
| | | SP | SP-AL | GS | GS-AL | CP | CP-AL | FN | FN-AL |
| Vicuna-7B | Llama-68M | 0.240 | 0.614 | 0.462 | 0.662 | 0.366 | 0.561 | 0.512 | 0.626 |
| | Llama-160M | 0.302 | 0.592 | 0.536 | 0.691 | 0.405 | 0.665 | 0.576 | 0.650 |
| FLAN T5-XL | T5-eff.-small | 0.404 | 0.539 | 0.315 | 0.517 | 0.562 | 0.700 | 0.338 | 0.633 |
| | T5-eff.-base | 0.583 | 0.690 | 0.387 | 0.623 | 0.576 | 0.796 | 0.301 | 0.686 |
| FLAN-T5-XL | FLAN-T5-small | 0.689 | 0.730 | 0.630 | 0.676 | 0.796 | 0.821 | 0.535 | 0.740 |
| | FLAN-T5-base | 0.736 | 0.782 | 0.711 | 0.751 | 0.818 | 0.840 | 0.641 | 0.774 |

model. The server is willing to offer an aligned public model because the alignment does not harm the server's privacy but allows a more efficient service. The server can offer a better-aligned public model than the client could achieve, leading to a higher acceptance ratio. The server has better insights into the training datasets and has access to more computational resources. The server can select a nonsensitive dataset that closely resembles the private dataset and select public models with a larger size. This is already the case when the server publishes smaller versions of the private model, such as Gecko in the Palm2 series from Google (Ghahramani, 2023) and Qwen2.5 in the Qwen series from Alibaba Cloud (Aliyun, 2025). Our experiments show that public models from the same series as their corresponding private models produce more aligned tokens, making our approach more appealing to the secure GPT decoding.

## 5. Experimental Results

Section 5.1 begins by detailing the experimental setup. Section 5.2 then presents the improvement in the acceptance ratio of draft tokens achieved through alignment and Section 5.3 highlights the improvement in end-to-end latency. Finally, Section 5.4 demonstrates the minimal overhead introduced by the optimized speculative sampling protocol. We omit the accuracy analysis (e.g., perplexity) since our method guarantees an identical output distribution to the private model, ensuring zero accuracy degradation.

### 5.1. Experimental Setup

**Models and Tasks** Throughout our experiments, we employ three pairs of private and public models, each differing in architectures and training hyperparameters. The first pair is Vicuna-7B (Chiang et al., 2023) and LLaMA-68M&160M (Miao et al., 2024a). Vicuna-7B is pre-trained on the CommonCrawl dataset (Wenzek et al., 2020) and fine-tuned using approximately 125,000 conversations from ShareGPT.com. LLaMA-68M&160M is pre-trained with the C4-en dataset (Raffel et al., 2020). The second pair is FLAN-T5-XL (3B) (Chung et al., 2024) and T5-efficient-small&base (Tay et al., 2021). FLAN-T5-XL is pre-trained

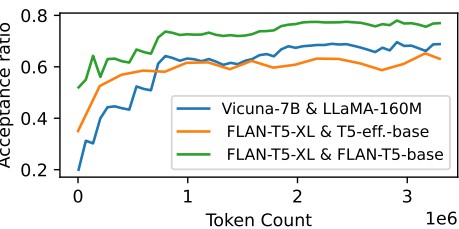

*Figure 4.* The alignment efficiency of three pairs of models on the Spider task.

on the C4 and Wiki-DPR datasets (Karpukhin et al., 2020), followed by fine-tuning on a wide range of downstream tasks across various languages. The T5-efficient series is only pre-trained on the C4 dataset without fine-tuning. The third pair examines the acceptance ratio when the server releases a small version of the private model from the same series. In this case, we compare FLAN-T5-XL (3B) with FLAN-T5-small&base. Among three pairs of models, we use models that from different series to show the robustness and general applicability and models from same series to show the performance in favorable setting.

We evaluate performance across four diverse tasks: Text-to-SQL (Spider) (Yu et al., 2018), graduate school math (Gsm8k) (Cobbe et al., 2021), Python code generation (Code-search-Python) (Husain et al., 2019), financial question answering (Alpaca-finance) (Gaurang Bharti, 2024).

**Secure Inference Setup** Performance evaluations are conducted on two nodes with 64 vCPUs and 128 GB memory. We utilize Linux Traffic Control (tc) to simulate Local Area Network (LAN) and Wide Area Network (WAN) environments, setting bandwidth and one-way delay to (1 Gbps, 10 ms) for LAN and (400 Mbps, 40 ms) for WAN.

**Baselines** To the best of our knowledge, prior works mainly optimize the protocols or modify the model architectures. POST is complementary to these works and can be integrated for further performance improvements. The performance

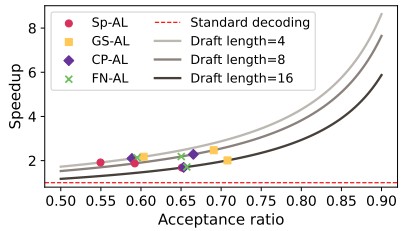
(a) Vicuna-7B & LLaMA-160M on LAN.

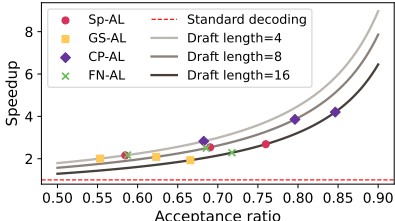
(b) FLAN-T5-XL & T5-eff.-base on LAN.

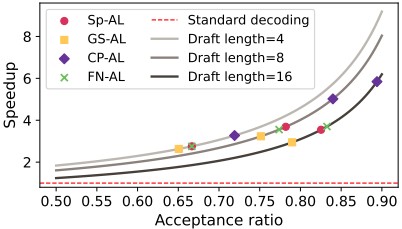
(c) FLAN-T5-XL & FLAN-T5-base on LAN.

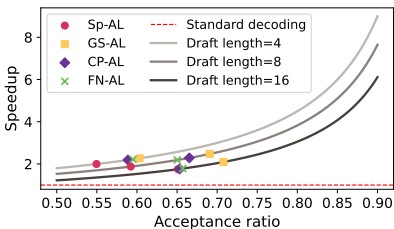
(d) Vicuna-7B & LLaMA-160M on WAN.

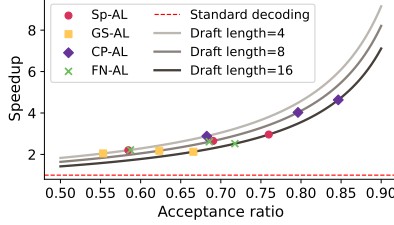
(e) FLAN-T5-XL & T5-eff.-base on WAN.

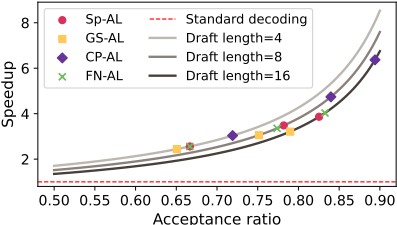
(f) FLAN-T5-XL & FLAN-T5-base on WAN.

*Figure 5.* The end-to-end speedup across two network settings and three pairs of models. The curves illustrate the relationship between speedup and acceptance ratio for various draft lengths. Specific speedups for four selected tasks are marked on these curves.

improvements are compared with SOTA inference protocols for the Transformer model, BumbleBee (Lu et al., 2025) and Nimbus (Li et al., 2024b). Specifically, we choose the linear protocol of Nimbus and the nonlinear protocol of BumbleBee, both of which have no negative impact on the model accuracy. All experiments are conducted on the well-established framework SecretFlow-SPU (Ma et al., 2023) for secure inference.

### 5.2. Alignment Results

**Improvements on Acceptance Ratios** Table 1 lists the acceptance ratios before and after alignment (AL). The results are based on eight draft tokens per decoding step. The acceptance ratio $\alpha$ represents the expected probability that a draft token is accepted. Consequently, our approach reduces the number of decoding steps by $\frac{1}{1-\alpha}$ compared to standard decoding. Across various model pairs and tasks, the acceptance ratios after alignment range from 52% to 84%, resulting in a $2.0\times$ to $6.3\times$ reduction in decoding steps compared to standard decoding.

The knowledge distillation enhances acceptance ratios. For model pairs from different series (the first two pairs), alignment increases the acceptance ratio from 10% to 40%, with final values ranging from 60% to 80%. For model pairs from the same series (the third pair), the improvement is approximately 10%, as the original public model already exhibits a high acceptance ratio, resulting in final acceptance ratios between 75% and 85%. Moreover, the results

indicate that larger public models achieve higher acceptance ratios, aligning with the intuition that larger models have greater capacity. In this study, the public GPTs employed are small and can be fine-tuned on a single GPU with 20 GB of memory. We anticipate that using larger public GPTs would further enhance the acceptance ratio.

**Alignment Efficiency** Figure 4 presents the relationship between the acceptance ratio and the number of tokens used for knowledge distillation. We use the spider task as an example and the similar trend also holds for other tasks. The results indicate that the alignment process is notably efficient regarding the token number. The acceptance ratios increase rapidly for all three model pairs, achieving a satisfying acceptance ratio after approximately 1 million tokens. Existing API of GPT service usually charges the client per token. Given the current API pricing (OpenAI, 2025), the total cost for this amount of tokens is approximately $10, demonstrating the practical feasibility. Moreover, as discussed in Section 4.3, a more effective approach is to let the server to perform the alignment.

### 5.3. End-to-end Performance

Figure 5 shows the speedup against the acceptance ratio using 4, 8, and 16 draft lengths. Markers on the curve indicate the speedup for four tasks. We present results for the selected model pairs and two network conditions. We also show the baseline for standard secure decoding (the red dot line), which has a speedup number one. For any task,

*Table 2.* Comparison of the naive and optimized secure speculative sampling.

| Network Condition | Draft Length | Decoder Time/s | Naive Time/s | Ours Time/s |
|---|---|---|---|---|
| 1 Gbps 10 ms | 4 | 7.11 | 14.78 | 1.19 |
|  | 8 | 8.67 | 28.26 | 1.45 |
| 400 Mbps 40 ms | 4 | 20.32 | 24.44 | 3.04 |
|  | 8 | 22.49 | 46.62 | 4.12 |

the client and server can choose the proper draft length for the best speedup, which we find ranges from $2.1\times$ to $6.0\times$.

Typically, greater draft lengths correspond to lower curves. Since the time spent on the public decoding and secure sampling is negligible, the primary reason is the increased decoding time associated with a larger input size. However, because a higher draft length leads to a higher expected number of accepted tokens per step, the overall end-to-end speedup may still be greater. Furthermore, the speedup is consistent across the two network conditions tested and tasks. This consistency demonstrates the great applicability in diverse deployment scenarios.

### 5.4. Secure Speculative Sampling Performance

Table 2 presents the performance of secure speculative sampling using Vicuna-7B as an example, both with and without optimization. For reference, we also report the secure evaluation time of a single decoder. The vocabulary size associated with the sampling time is 32000, which is similar for different private models. Without optimization, the sampling process incurs a latency equivalent to the latency of two decoders, which noticeably increases end-to-end delay. In contrast, our optimization yields approximately a ten-fold latency reduction, rendering its impact on end-to-end latency negligible.

## 6. Related Work

**Private Transformers.** One type of works are to optimize cryptographic protocols to improve the efficiency of the secure inference. For linear layers, existing works usually modify the encoding methods (Hao et al., 2022; Pang et al., 2024; Li et al., 2024b) and the ciphertext packing strategy (Lu et al., 2025; He et al., 2024) to improve the computation and communication. For nonlinear layers, Crypt-Flow2 (Rathee et al., 2020) proposes a faster millionaire protocol to reduce communication size. Other methods, such as look-up tables for faithful approximation (Rathee et al., 2021; Gupta et al., 2023; Pang et al., 2024), are computationally expensive to maintain model accuracy.

Another type of works is to modify the Transformer model to tailor the cryptographic primitives. Some works integrate

the sparsity to linear layers to reduce the homomorphic operations (Zimerman et al., 2024; Ran et al., 2023; Xu et al., 2024) For linear layers, some works (Chen et al., 2022; Zeng et al., 2023; Li et al., 2022) use aggressive approximation of Softmax and GELU. However, such aggressive approximations lead to noticeable accuracy loss, even when employing knowledge distillation to mitigate the decline in accuracy. A more accurate way is to use piecewise polynomial to approximate the nonlinear layers (Dong et al., 2023; Lu et al., 2025; Li et al., 2024b). They use high-degree polynomial and multiple pieces. It does not result in an accuracy drop but is also relatively costly to compute.

In contrast, this work examines a different aspect by leveraging the knowledge shared between public and private models to accelerate secure inference, which can be combined with prior work for further speedup.

**Speculative Decoding** Speculative decoding is a recent technique designed to enhance the efficiency of autoregressive decoding of the GPTs (Leviathan et al., 2023; Cai et al., 2023; Liu et al., 2023). In the original speculative decoding, several draft tokens are generated using a tiny GPT and are forwarded by a large GPT in parallel, which also inspires the proposed POST approach. However, POST differs from the original speculative decoding in three key aspects. First, POST is based on a unique observation that cryptographic protocols in secure GPT decoding demonstrate insensitivity to input length. Second, speculative decoding must carefully balance the computational costs between the tiny and large GPT models. In contrast, in this work, the cost associated with the public model is negligible compared to the secure decoding process of the private model, and we primarily focus on optimizing the cost of securely executing the speculative sampling algorithm. Third, while speculative decoding can freely choose models to propose draft tokens, this work selects public models that are not closely related to the private model. Consequently, we employ model alignment to demonstrate that pairs of models with low relevance can still achieve a high acceptance ratio following knowledge distillation.

## 7. Conclusion

This work proposes a novel POST approach to secure GPT inference. Our approach utilizes knowledge already disclosed by public models for acceleration while maintaining the same privacy goal and generation quality comparable to standard secure decoding. Our approach demonstrates a speedup ranging from $2.1\times$ to $6.0\times$ across various settings. Furthermore, our approach offers the potential for even greater speedup when applied to better-aligned public models, such as those provided by servers or larger public models. These optimizations enhance performance, advancing the practical application of secure GPT inference.

## Acknowledgement

This work was supported by the National Natural Science Foundation of China grants (62222210, 62125204, and 92270201). This work was also supported by Shanghai Qi Zhi Institute Innovation Program SQZ202316. Corresponding authors are Jingwen Leng (leng-jw@sjtu.edu.cn), Kang Yang (yangk@sklc.org), and Yu Yu (yyuu@sjtu.edu.cn).

## Impact Statement

This paper presents work whose goal is to advance the privacy-preserving GPT inference. Our work introduces a more efficient method for privacy inference, which contributes to providing active protection against potential privacy issues arising in the application of GPT.

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

# A. Background of the Generative Pre-trained Transformer (GPT)

We focus on GPTs (Vaswani et al., 2017), such as Vicuna (Chiang et al., 2023) and FLAN-T5 (Chung et al., 2024). These models are stacked with Transformer decoders, each consisting of an attention module and a feed-forward network (FFN) module.

**Attention Module.** The attention module starts with three independent linear layers $\mathsf{Linear}_{qkv}$ that project the input to three activation tensors: $\mathbf{Q}$, $\mathbf{K}$, and $\mathbf{V}$. The multi-head attention mechanism splits them into and computes the self-attention of all heads in parallel. The $h_{th}$ head is computed through

$$\mathbf{Attention}(Q^h, K^h, V^h) = \mathsf{Softmax}\left(\frac{Q^h K^{h^T}}{\sqrt{d_k}}\right) V^h, \tag{3}$$

where $d_k$ is the hidden dimension of the key activation. The outputs of different heads are concatenated and fed into another linear layer $\mathsf{Linear}_o$, with one residue connection and one normalization layer to generate the final output of the attention module.

**FFN Module.** The FFN module is composed of two linear layers $\mathsf{Linear}_{h1}$ and $\mathsf{Linear}_{h2}$, and one activation layer ACT. The FFN module is computed as follows

$$\mathbf{FFN}(\mathbf{X}) = \mathsf{Linear}_{h_2}(\mathsf{ACT}(\mathsf{Linear}_{h_1}(\mathbf{X}))), \tag{4}$$

where ACT can be many variants, such as ReLU, GELU, and SiLU Similar to the attention module, its output needs a residue connection and a normalization layer.

**Embedding Layer.** The embedding layer is positioned at the beginning of the GPT. In this layer, each token is mapped from its index in the vocabulary to a fixed-size vector. Given that transformers do not possess inherent sequence awareness, positional encodings are incorporated into the token embeddings to convey information regarding the order of tokens within the sequence.

**Sampling.** The final task head in GPT predicts the output distribution of the subsequent token given an input prefix. This task head generally comprises a linear layer that projects the hidden states from the final transformer decoder to the dimensionality of the vocabulary, along with a $\mathsf{Softmax}$ function that normalizes the resulting probability distribution. Then, the next token is sampled from the distribution. While various sampling methods exist, such as top-k and temperature settings, they can all be conceptualized as standard sampling from an adjusted original probability distribution. For instance, argmax sampling is equivalent to zeroing out all non-maximal elements of the distribution and normalizing the result. Thus, we can concentrate exclusively on standard sampling from a probability distribution.

**Prefill and Decoding Phase.** The prefill phase begins text generation when GPT processes the initial input prompt. All input tokens are fed into the GPT during the prefill phase. The model generates a comprehensive set of probabilities for each input token. However, in both the prefill and subsequent decoding phases, only the final output distribution is of interest, as this represents the likelihood of generating the next token based on the provided input context. Consequently, the prefill phase generates the output distribution for the subsequent token and initializes the key-value (KV) cache. The KV cache prevents re-computing the representations of all previous tokens when decoding a new token. It stores the keys and values associated with earlier tokens, thereby preserving the contextual information required for generating the following token.

The decoding phase generates tokens sequentially using the context from the prefill and previously generated tokens. With the presence of the KV cache, the model computes the output distribution for the new token at each decoding step and updates the information within the KV cache accordingly.

# B. Complete POST Protocols

This section gives the full version of the POST approach, including the process of public decoding, secure verification, and the re-sampling of the "bonus" token.

---

**Algorithm 2** Full Algorithm of one decoding step using POST

---

**Input:** $P_s$ (server) holds private model $\mathcal{M}_p$ and generates . $P_c$ (client) holds prefix $x_{<t}$ and public model $\mathcal{M}_q$. Public parameters include draft length $\gamma$. The vocabulary size is $\mathcal{V}$. The $[\gamma]$ means $[0, 1, \ldots, \gamma]$

1: **for** $i \in [\gamma]$, $P_c$ **do**
2:      $q(x \mid x_{x<t+i}) = \mathcal{M}_q(prefix + [x_1, \ldots, x_{i-1}])$.
3:      $x_i \sim q(x \mid x_{x<t+i})$.
4: **end for**
5: $P_s$ and $P_c$ parallely compute $\langle p(x \mid x_{<t}) \rangle, \cdots, \langle p(x \mid x_{<t+\gamma}) \rangle = \mathcal{M}_p(\langle prefix \rangle), \ldots, \mathcal{M}_p(\langle prefix + [x_1, \ldots, x_\gamma] \rangle)$.
6: $P_c$ sets $\mathbf{Q} = \begin{bmatrix} q(x \mid x_{<t}) \\ \cdots \\ q(x \mid x_{<t+\gamma-1}) \end{bmatrix}$. $P_s$ and $P_c$ set $\langle \mathbf{P} \rangle = \begin{bmatrix} \langle p(x \mid x_{<t}) \rangle \\ \cdots \\ \langle p(x \mid x_{<t+\gamma}) \rangle \end{bmatrix}$.
7: $P_c$ generates a random vector $\mathbf{R}_{mul} \sim U(0, 1)^{\gamma \times \mathcal{V}}$.
8: $P_s$ sets his share of $\langle \mathbf{S} \rangle_s = -\langle \mathbf{P} \rangle_s$ and $P_c$ sets her share $\langle \mathbf{S} \rangle_c = \mathbf{Q} \cdot \mathbf{R}_{mul} - \langle \mathbf{P} \rangle_c \mod 2^\ell$.
9: **for** $i \in [\gamma]$, in parallel **do**
10:      $P_s$ generates a random vector $\mathbf{r} \sim \mathbb{Z}_{2^\ell}^\gamma$, and masks his share as $\langle \hat{\mathbf{S}}[i, :] \rangle_s = \langle \mathbf{S}[i, :] \rangle_s - \mathbf{r}[i] \mod 2^\ell$.
11:      $P_c$ chooses the secret share by $\langle \hat{\mathbf{S}}[i, x_i] \rangle_s \sim \binom{\mathcal{V}}{1} - \text{OT}_\ell(\langle \hat{\mathbf{S}}[i, :] \rangle_s, x_i)$.
12:      $P_s$ sets his share $\langle \mathbf{s} \rangle_s[i] = \mathbf{r}[i]$ and $P_c$ sets her share $\langle \mathbf{s} \rangle_c[i] = \langle \mathbf{S}[i, x_i] \rangle_c + \langle \hat{\mathbf{S}}[i, x_i] \rangle_s \mod 2^\ell$.
13: **end for**
14: $P_s$ and $P_c$ compute shares of boolean vector $\langle \mathbf{n} \rangle = \mathcal{F}_{less}(0, \langle \mathbf{s} \rangle)$ and open $\mathbf{n}$ to $P_c$.
15: $P_c$ appends an element one at the end by $\mathbf{n} = \mathbf{n} + [1]$
16: $P_c$ computes $k = \min(\{i \mid i \in [\gamma], \mathbf{n}[i] = 1\})$.
17: $P_c$ chooses $\langle p_k(x) \rangle_s \sim \binom{\gamma+1}{1} - \text{OT}_\ell([\langle p(x \mid x_{<t}) \rangle_s, \cdots, \langle p(x \mid x_{<t+\gamma}) \rangle_s], k)$.
18: $P_c$ reconstructs $p_k(x) = \langle p_k(x) \rangle_s + \langle p_k(x) \rangle_c$.
19: **if** $k < \gamma$ **then**
20:      $P_c$ computes $p'(x) = \max(0, p_k(x) - q_k(x))$.
21: **else if** $k == \gamma$ **then**
22:      $P_c$ sets $p'(x) = p_{\gamma+1}(x)$.
23: **end if**
24: $P_c$ samples $x_k \sim p'(x)$.
**Output:** $P_c$ obtains $prefix + [x_1, \ldots, x_k]$.

---

## C. Correctness of the Secure Sampling Protocol

Speculative sampling samples token from private model's output distribution $p(x)$ and public model's output distribution $q(x)$. In the original algorithm, the token proposed by the public model is rejected according to the following probability

$$\max(0, 1 - \frac{p(\hat{x})}{q(\hat{x})}). \tag{5}$$

where $\hat{x}$ is the token index sampled from the output distribution of the public model. Next, we show that the protocol exactly follows the original functionality.

We first show the correctness of refactoring division to multiplication. The original rejection is to sample a uniform random

number $r \in [0, 1]$, and reject token if $r < \max(0, 1 - \frac{p(\hat{x})}{q(\hat{x})})$. An equivalent way is to compute

$$
\begin{aligned}
r &< \max\left(0, 1 - \frac{p(\hat{x})}{q(\hat{x})}\right) \\
1 - r &\geq \min\left(1, \frac{p(\hat{x})}{q(\hat{x})}\right) \\
r &\geq \min\left(1, \frac{p(\hat{x})}{q(\hat{x})}\right) \\
r &\geq \frac{p(\hat{x})}{q(\hat{x})} \\
r * q(\hat{x}) &\geq p(\hat{x}).
\end{aligned}
\tag{6}
$$

The equivalence of second line and third line is because $r$ and $1 - r$ follow the same distribution when $r \in [0, 1]$. The equivalence of the conditions in line 3 and line 4 is established through two cases based on the range of $\frac{p(\hat{x})}{q(\hat{x})}$. If $\frac{p(\hat{x})}{q(\hat{x})} \in [0, 1]$, the $min()$ function can be directly removed; if $\frac{p(\hat{x})}{q(\hat{x})} \in (1, \infty)$, the third line simplifies to $r > 1$. Since $r$ is drawn from a uniform distribution over the interval $[0, 1]$, the conditions in both line 3 and line 4 are never satisfied. Consequently, the equivalence remains valid. The final line shows the correctness of the refactoring regarding one draft token. The correctness of the protocol is obtained easily by extending the result to all proposed tokens.

For the selection then comparison, the correctness directly follows the original algorithm that computes the comparison results of the target element in the score matrix $\mathbf{S}$.

## D. Security Analysis of the Secure Sampling Protocol

This section proves that secure inference on POST maintains the same privacy goal as the standard GPT decoding, i.e., the client only learns the model's output distribution, and the server learns nothing. The security analysis can be categorized into two parts: security within the MPC protocol and security outside the MPC protocol.

- For the data processed using MPC protocol, both the client and the server learn nothing. Our protocol calls the underlying cryptographic protocols a black box, which is the same as the common practice of secure inference works. Such black-box usage of the cryptographic protocols makes security straightforward and achieved according to the composability of cryptographic protocols.

  A tricky part in our protocol is that we use the same random ring number to mask one row of the score matrix $\mathbf{S}$. This is not secure in other cases as $\mathbf{S}[i, :] - r \mod 2^{\ell}$ makes the relative difference leaked. Fortunately, the client in our protocol only learns only one element $\mathbf{S}[i, x_i] - r \mod 2^{\ell}$ and prevents such leakage.

- For public values outside MPC, we show they do not reveal additional information than standard GPT inference. First, there is nothing else to prove for the client's security since the server does not receive any messages in our protocol. Second, to prove the server's security, we compare the information revealed to the client in the standard decoding and POST. We first give some notations. In the standard secure GPT decoding, for a prefix $\mathbf{x}_{<t}$, the client can autoregressively obtain $\gamma$ output distributions $p(x \mid x_{<t}), \ldots, p(x \mid x_{<t+\gamma})$ of the private model, from which the client samples tokens $x_1, \ldots, x_\gamma$. In the POST, the client learns the rejection boolean of all draft tokens and the private model's output distributions $p(x \mid x_{<t+k})$ of the first rejected token. The output distribution is the final output, which is also revealed to the client in the standard decoding. The indexes of rejected tokens are a direct result when the client learns which token is generated in the standard decoding. For example, for the $i_{th}$ generated token, the client can always know that the proposing draft token $x_i$ will receive acceptance while any other token will receive rejection.

  A more beneficial result is that POST only lets the client obtain a subset of information than the standard secure decoding. The client only obtains output distributions $p(x \mid x_{<t+k})$. For other accepted tokens, it is equivalent that the client only learns the Top1 element of the corresponding output distribution. This is in contrast to the standard decoding, where the client can learn the output distribution's TopK elements (usually K=5). In this way, the client information of POST is a subset of those in the standard decoding.

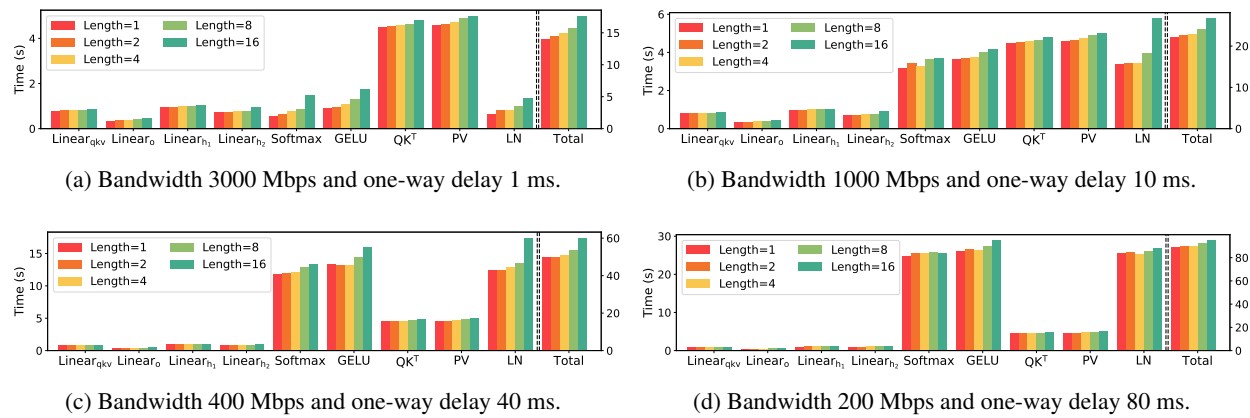

(a) Bandwidth 3000 Mbps and one-way delay 1 ms.

(b) Bandwidth 1000 Mbps and one-way delay 10 ms.

(c) Bandwidth 400 Mbps and one-way delay 40 ms.

(d) Bandwidth 200 Mbps and one-way delay 80 ms.

*Figure 6.* The latency of each layer against different input input lengths. We show the results of GPT-2 on EzPC frameworks.

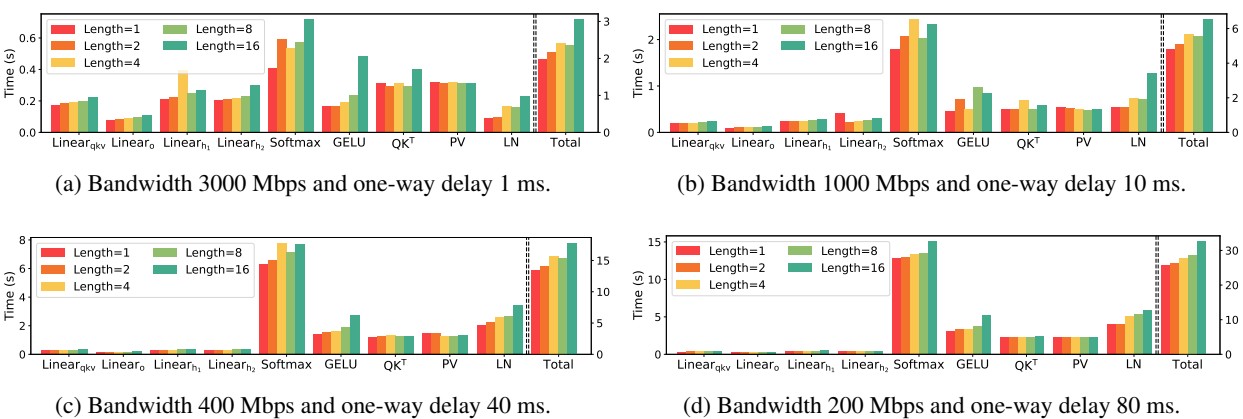

(a) Bandwidth 3000 Mbps and one-way delay 1 ms.

(b) Bandwidth 1000 Mbps and one-way delay 10 ms.

(c) Bandwidth 400 Mbps and one-way delay 40 ms.

(d) Bandwidth 200 Mbps and one-way delay 80 ms.

*Figure 7.* The latency of each layer against different input input lengths. We show the results of GPT-2 on SecretFlow-SPU frameworks.

# E. More Experiments

## E.1. Latency Insensitivity to the Input Length

For the decoding of GPT, the input length of the operators is only 1. Section 3.2 observes that the latency of cryptographic protocols is insensitive to input length at such minimal input length. This section presents additional results to further substantiate the observation. Our experiments encompass the following aspects.

- Network Conditions: We employ four network conditions to simulate different Local Area Network (LAN) and Wide Area Network (WAN) scenarios, specifically varying bandwidth and one-way delay. The configurations are as follows: (3000 Mbps, 1 ms), (1000 Mbps, 10 ms), (400 Mbps, 40 ms), and (200 Mbps, 80 ms).

- GPT Sizes: In secure inference studies, many existing works focus on models comparable in size to GPT-2. Since models of such size are the closest to the practical usage of secure inference, we also include the GPT-2 model to illustrate our observations. Additionally, we incorporate two larger language models: FLAN-T5-XL (3 billion parameters) and Vicuna-7B.

- Secure Inference Frameworks: We leverage two well-established frameworks: EzPC (Chandran et al., 2017) and SecretFlow-SPU (Ma et al., 2023). These frameworks utilize different underlying primitives and protocols for both linear and nonlinear layers.

  For the EzPC framework, we adopt the state-of-the-art work BOLT (Pang et al., 2024), which is implemented in EzPC, to profile the layers. The HE protocol employed in BOLT implements matrix multiplication through Number

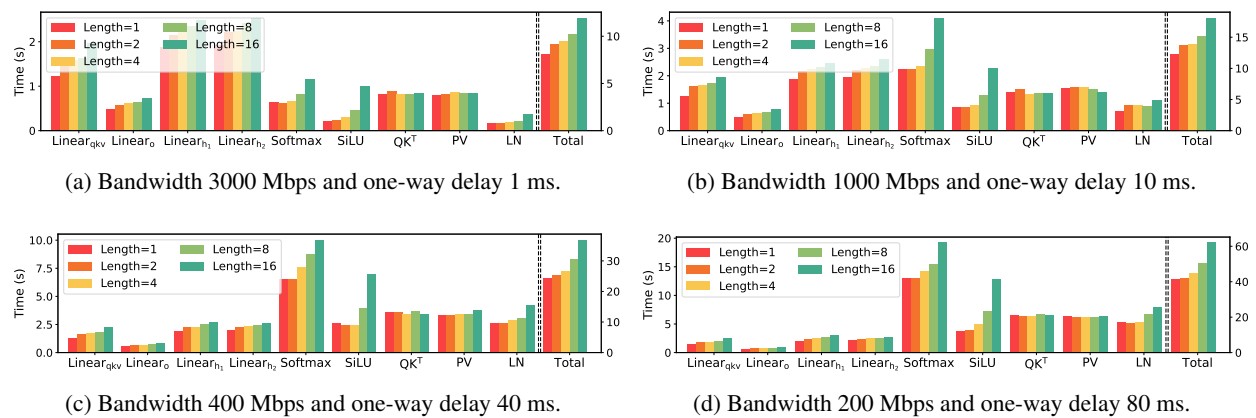

(a) Bandwidth 3000 Mbps and one-way delay 1 ms.

(b) Bandwidth 1000 Mbps and one-way delay 10 ms.

(c) Bandwidth 400 Mbps and one-way delay 40 ms.

(d) Bandwidth 200 Mbps and one-way delay 80 ms.

*Figure 8.* The latency of each layer against different input input lengths. We show the results of Vicuna-7B on SecretFlow-SPU frameworks.

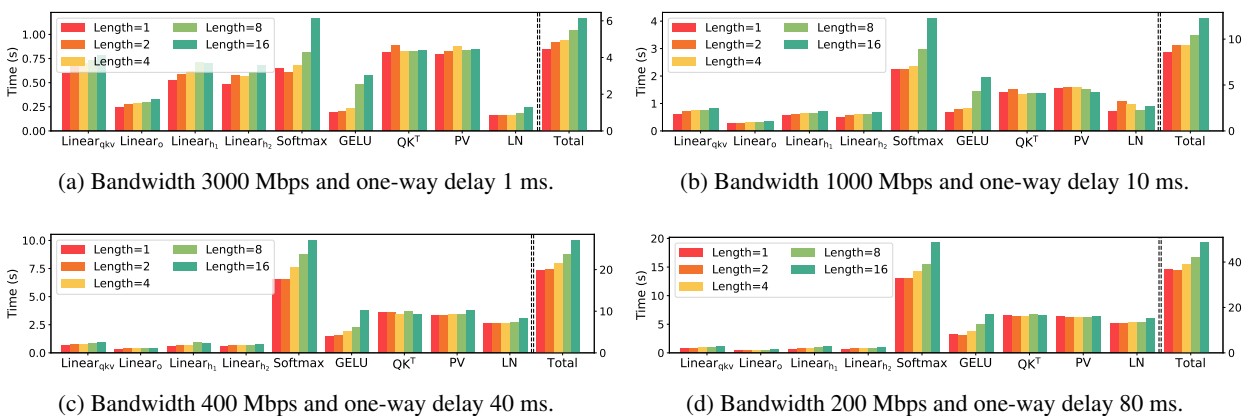

(a) Bandwidth 3000 Mbps and one-way delay 1 ms.

(b) Bandwidth 1000 Mbps and one-way delay 10 ms.

(c) Bandwidth 400 Mbps and one-way delay 40 ms.

(d) Bandwidth 200 Mbps and one-way delay 80 ms.

*Figure 9.* The latency of each layer against different input input lengths. We show the results of FLAN-T5-XL on SecretFlow-SPU frameworks.

Theoretic Transform (NTT) encoding. Both polynomial approximation and lookup tables are utilized to evaluate the nonlinear layers. The underlying OT protocol (Kolesnikov & Kumaresan, 2013) is an enhancement of the IKNP-OT protocol (Ishai et al., 2003). We only present profiling results for GPT-2 within the EzPC framework, as BOLT's implementation does not support larger model sizes.

For the SecretFlow-SPU framework, we utilize the state-of-the-art works BumbleBee (Lu et al., 2025) and Nimbus (Li et al., 2024b) to conduct our experiments. They employ coefficient encoding for linear layers and piecewise polynomial functions for nonlinear layers. The underlying OT protocol in SecretFlow-SPU is the Ferret-OT (Yang et al., 2020).

Across various settings, we observe a consistent insensitivity of latency to input length. When comparing the latency for input lengths of 16 and 1, the observed ratio ranges from $1.07\times \sim 1.77\times$. Next, we give a detailed analysis from different perspectives.

In examining four distinct network conditions, we note that latency increases more noticeably under optimal network conditions. For example, when comparing the latency for input lengths of 1 and 16 at a bandwidth of 3000 Mbps and a latency of 1 ms, the increasing ratio varies from $1.26\times \sim 1.77\times$. In the case of 1000 Mbps and 10 ms, the increasing ratio ranges from $1.21\times \sim 1.55\times$. For 400 Mbps and 40 ms, the ratio ranges from $1.21\times \sim 1.43\times$, while for 200 Mbps and 80 ms, it is $1.07\times \sim 1.40\times$. The reason can be attributed to the less significant one-way delay under good network conditions. Given that one-way delay is completely invariant to input length, its reduction will render overall latency more sensitive to input length variations.

When comparing different frameworks, we find that the EzPC framework exhibits greater insensitivity than the SecretFlow-SPU framework. This discrepancy is partly attributable to the implementation methods of the cryptographic protocols. The SecretFlow-SPU sometimes splits data and overlaps communication with computation. We find that an increase in input length may necessitate additional rounds of communication, which consequently increases one-way delay time about input size. In contrast, the EzPC framework maintains constant communication rounds regardless of input length.

Regarding different model sizes, smaller models demonstrate greater insensitivity compared to larger models. For instance, in the case of GPT-2, both the EzPC and SecretFlow-SPU frameworks exhibit nearly unchanged layer latencies. This phenomenon is because smaller models have smaller hidden dimensions, requiring greater input lengths to produce a noticeable increase in latency. However, the Softmax layer is an exception, as it is applied on a tensor whose size relates solely to the KV cache size.

### E.2. Detailed Analysis for Different Layers

We also provide a detailed analysis of the different layers. The layers can be generally categorized into linear and nonlinear layers.

- **Linear Layers.** We first discuss the matrix multiplication between the plaintext weights and ciphertext activations, including $\mathsf{Linear_{qkv}}$, $\mathsf{Linear_o}$, $\mathsf{Linear_{h1}}$, and $\mathsf{Linear_{h2}}$. The required operation includes one matrix multiplication and one truncation. Overall, the most cost is spent on the homomorphic operation, including the encryption, decryption, and multiplication of the polynomial ring. As we have explained in Section 3.2, the computation and communication efficiency of the SIMD HE operations heavily depends on the number of activations. The typical degree 8192 of the polynomial ring allows 8192 values to be operated in parallel. However, the single activation vector in the standard decoding is insufficient to fully utilize the slots in the polynomial ring, especially for the small model GPT-2. Existing works typically adopt packing methods to improve the utilization ratio of the slots. For example, the BOLT packs multiple input activation vectors into one plaintext polynomial. The polynomial is then encrypted and sent to the server for computation. In the Nimbus, the output ciphertext is packed into one and re-shared between parties. Therefore, when increasing the input length, the computation and communication efficiency can be improved, leading to sublinearly increased communication and computation times.

  Similar rules also hold for the matrix multiplication between activations, including $\mathsf{QK^T}$ and $\mathsf{PV}$. Consider the multiplication between secret matrices $\mathbf{X}$ and $\mathbf{Y}$, it is computed by $\mathbf{XY} = \langle\mathbf{X}\rangle_c\langle\mathbf{Y}\rangle_c + \langle\mathbf{X}\rangle_c\langle\mathbf{Y}\rangle_s + \langle\mathbf{X}\rangle_s\langle\mathbf{Y}\rangle_c + \langle\mathbf{X}\rangle_s\langle\mathbf{Y}\rangle_s$. The problem is transformed into two instances of plaintext-ciphertext multiplications of the two cross terms, which is the problem mentioned in the above paragraph. The low utilization of the SIMD operation is more significant for the activation matrix multiplication due to the multi-head mechanism. The hidden dimension of the matrix multiplication is further divided by the head number, leading to a lower utilization ratio and minimal sensitivity to the input length.

- **Nonlinear Layers.** The nonlinear layers, including Softmax, GELU, SiLU, and layernorm. These layers are usually approximated by the piecewise polynomials that only include multiplication and comparison. Multiple truncations are also called to reset the fixed-point precision after the multiplication. These nonlinear operations require numerous rounds of communication and transmitted data and are the main source of latency. Since the time spent on the one-way delay is fixed despite the input length, when the input size is small, the time spent on the transmission is less significant than the one-way delay. This is much more obvious for the layernorm, as the reciprocal square root is only computed on one secret number for a whole vector. Almost all latency is spent on the one-way delay. In this way, we observe the insensitivity of the overall latency.

