# OpenReview forum: "An Efficient Private GPT Never Autoregressively Decodes"
_ICML.cc/2025/Conference — ICML 2025 poster_

### Official Review · Reviewer_xivh · 2025-03-10

**Overall Recommendation:** 3

**Summary:**

The authors mainly aim to improve the efficiency of private inference for autoregressive language models. First, they observe that the decoding time is relatively insensitive to the input length. Next, they adapt speculative decoding to the private inference setting. The authors employ a small public model as a drafter and a larger private model as a verifier. A key technical challenge lies in the verifier's rejection rule, which typically requires computationally expensive reciprocal operations. To address this, the authors propose a novel protocol that allows rejection of draft tokens. Experimental results demonstrate improvements in decoding speed, achieving 2.1× to 6.0× efficiency gains.

## update after rebuttal
I appreciate the authors for commenting on the remaining concern. Using a smaller public model may be partly due to resource constraints, but the main reason is the copyright for larger models. If a public model could match the performance of a private one, there would be no need to use private models. Therefore, in realistic scenarios, public models are inherently expected to underperform compared to private models. As a result, for complex tasks like reasoning, the accuracy gap is likely to remain significant. While the proposed method may offer some speedup in such tasks, the overall impact would likely be limited. For this reason, I believe the broad effect of the work is not substantial enough and will maintain my score as weak accept.

**Claims And Evidence:**

* They claim that the decoding time is relatively insensitive to the input length based on Figure 1 and Figure 2. However, for softmax, their computation cost increases drastically as the input length increases in Figure 1. Thus, if the input length is over 64 or 128, the input length might be one of the major bottlenecks. The comparison when input length is 64 would make the claim stronger.

* The efficiency of the proposed protocol for rejection rule is shown in Table 2.

**Essential References Not Discussed:**

None

**Experimental Designs Or Analyses:**

See Evaluation Criteria.

**Methods And Evaluation Criteria:**

* Efficiency evaluation is well presented in Figure 5.

* The paper does not report model performance (e.g., accuracy or perplexity), focusing solely on speed improvements.  While speculative decoding with hard matching theoretically preserves model outputs, the use of soft matching may introduce performance degradation. An empirical analysis of how soft matching impacts accuracy or output quality would be valuable for practitioners, especially to understand the trade-off between efficiency and performance.

**Other Comments Or Suggestions:**

None.

**Other Strengths And Weaknesses:**

* In practical deployments, the drafter is constrained to be a small LLM. However, small models typically lack the capacity to generalize across diverse tasks and require task-specific alignment. Obtaining aligned drafters for every possible task is impractical, and without proper alignment, the effectiveness of speculative decoding diminishes. This limitation is evident in Table 1, where the rejection rate increases substantially when the drafter is not aligned with the target task.

**Questions For Authors:**

See the above sections.

**Relation To Broader Scientific Literature:**

* If performance is preserved, the paper would be significant, as it demonstrates substantial speedups by leveraging public drafter models for private inference.

**Theoretical Claims:**

There are no theoretical claims.

---

> ### Author Rebuttal · Authors · 2025-03-30
>
> Thank you for the positive feedback and appreciation of our work. We appreciate your insights and would like to provide more clarification.
> # Question 1: Figure 1&2 claim that the decoding time is relatively insensitive to the input length. However, for softmax, the cost increases drastically as the input length increases. Thus, if the input length is over 64 or 128, its time might be one of the major bottlenecks.
> We appreciate the reviewer's rigorous feedback and will enhance our demonstration to clarify this claim. In standard decoding, the input length is fixed at one. Our method's selection of the input (or draft) length should keep the latency insensitivity. Experiments show that using too many draft tokens (e.g., 64 or 128) indeed increases latency, so we avoid such large numbers. Instead, we focus on a reasonable range, like 1 to 16 draft tokens in Figure 1, or 4, 8, and 16 in Figure 5.
>
>
> # Question 2: If performance (perplexity) is preserved, the paper would be significant, as it demonstrates substantial speedups by leveraging public drafter models for private inference.
> We would like to clarify that soft matching through speculative sampling theoretically preserves the accuracy performance (e.g. perplexity) of the private model [1], as we mentioned in Line 239. It guarantees that tokens sampled from the distributions $p(x)$ and $q(x)$ using speculative sampling are distributed identically to those sampled from $p(x)$ alone. This implies that the probability of sampling any token $\hat{x}$ from both $p(x)$ and $q(x)$ is precisely $p(\hat{x})$. As a result, the expected perplexity of the sampled tokens with respect to some oracle baseline remains unchanged, thereby maintaining the performance concerned by the reviewer.
>
> We provide a brief proof of why the tokens sampled by two types of sampling algorithm follow an identical distribution; further details can be found in Appendix A of [1]. The probability of sampling a token $\hat{x}$ using speculative sampling is:
> $$
> P\\{x = \hat{x}\\} = P\\{x = \hat{x} | acc\\} \cdot P\\{acc\\} + P\\{x = \hat{x} | rej\\} \cdot P\\{rej\\}
> $$
> - When the proposed token is accepted: The public model proposes $\hat{x}$ with probability $P\\{x = \hat{x} | acc\\} = q(\hat{x})$. The acceptance probability for $\hat{x}$ is $P\\{acc\\} = \min\left(1, \frac{p(\hat{x})}{q(\hat{x})}\right)$.
> - When the proposed token is rejected: The public model may propose any token, which is then rejected. The rejection probability is $P\\{rej\\} = \sum_{x'} q(x') \cdot \left(1 - \min\left(1, \frac{p(x')}{q(x')}\right)\right) = 1 - \sum_{x'} \min(q(x'), p(x'))$. The token is re-sampled from the adjusted distribution with $P\\{x = \hat{x} | rej\\} = norm(\max(0, p(\hat{x}) - q(\hat{x})))$.
>
> Substituting these into the equation, we obtain:
> $$
> P\\{x = \hat{x}\\} = q(\hat{x}) \cdot \min\left(1, \frac{p(\hat{x})}{q(\hat{x})}\right) + \left(1 - \sum_{x'} \min(q(x'), p(x'))\right) \cdot norm(\max(0, p(\hat{x}) - q(\hat{x}))) = p(\hat{x})
> $$
> This confirms that the speculative sampling process preserves the original distribution $p(x)$, as desired. If you have any further questions, please feel free to let us know.
>
> [1] Fast inference from transformers via speculative decoding. ICML 2023
>
> # Question 3: The paper does not report model performance (e.g., perplexity). An empirical analysis of how soft matching degrades output quality would be valuable for practitioners.
> As discussed in Question 2, there is no inherent trade-off between efficiency and accuracy performance when adopting the soft maching. The soft matching theoretically maintains the accuracy performance, i.e. same expected perplexity towards some oracle output. This theoretical guarantee is a key strength of our method, allowing speed improvements without concerning the output quality.
>
> # Question 4: In practice, drafter models are limited to small LLMs, which often lack generalization across tasks and need task-specific alignment. Aligning drafters for every task is impractical, and without proper alignment, the efficiency declines.
> In our experiments, even using non-aligned small models, we achieve an average acceptance rate of 40% for the different-series model and an average of 65% for the same-series model, which already presents approximately 1.6X and 3X average speedup.
>
> The alignment further increases the benefits to 2X and 5X average speedup. Notably, the alignment is much easier and more practical than traditional fine-tuning tasks that prioritize accuracy as the primary objective. As illustrated in Figure 4, the alignment requires only a small public GPT, a minimal aligning dataset, and can be efficiently tuned on a small GPU.
>
> Furthermore, the effectiveness of our approach is expected to increase as client-side computational capabilities continue to advance. Such advancements enable the leveraging of larger and more powerful public models, thereby enhancing the acceptance ratio as Table 1 indicates.

---

> > ### Comment · Reviewer_xivh · 2025-04-05
> >
> > I appreciate the detailed response. Most of my concerns have been addressed. However, given the inherent limitations of small LLMs, the potential gains in broader applications such as reasoning remain limited. As a result, I am maintaining my score as a weak accept.

---

> > > ### Author Response · Authors · 2025-04-05
> > >
> > > We are glad that previous response addressed most of your concerns. Regarding your remaining question about limited capabilities of small public models, besides the evidence in prievious response, we offer a further detailed explanation about the choice of public models. In fact, in the secure inference scenario considered in this paper, clients can freely choose larger public models, which is precisely where the potential of our method lies.
> > >
> > > This flexibility in choosing public models arises from the inherent orders-of-magnitude latency gap between plaintext and secure inference [1,2], as we also explained in Line 220. For models of the same size and running in the same environment, secure inference is typically hundreds of times slower than plaintext inference. As a result, in our POST approach, compared to the major bottleneck of secure verification, the time spent on public decoding is a negligible part, as the following Table shows.
> > >
> > > **Table 1: The latencies for the public model autoregressively generate 8 tokens, and the latencies for the private model verify 8 tokens in one forward pass. All latencies use the same CPU environments. We use seconds to measure the latency.**
> > > ||Public decoding|Secure verification (3000Mbps, 1ms)|Secure verification (1000Mbps, 10ms)|
> > > |-|-|-|-|
> > > |GPT-2|0.14|27.6|67.2|
> > > |Vicuna-7B|3.36|320|480|
> > > |FLAN-T5-XL|1.82|132|238|
> > > |T5-efficient-XL|1.94|148|243|
> > >
> > > According to the above results, even if clients select a public model of the same size as the private model, the time spent on public decoding remains negligible in the end-to-end delay, while a higher acceptance ratio (speedup) is expected to be achieved. Therefore, the flexibility to choose larger models shows promising acceleration potential of our method.
> > >
> > > Additionally, a potentially overlooked aspect is that clients capable of performing secure inference typically have the resources to utilize larger LMs than those minimal LMs in our paper. As a price of the privacy protection, secure inference [3,4,5] requires clients to possess some computational and communication capabilities (which aligns with the trend of increasingly powerful client devices, such as those equipped with GPUs). Such demand is due to the widely used secure inference protocols that often require clients to perform computations and communications comparable to those of the server. For example, the computational and communication between two parties are nearly symmetric in MPC protocols, and clients perform heavy encryption and decryption in HE multiplication.
> > >
> > > [1] SecretFlow-SPU: A performant and User-Friendly framework for Privacy-Preserving machine learning, ATC 2023
> > >
> > > [2] https://github.com/mpc-msri/EzPC, 2024
> > >
> > > [3] Bumblebee: Secure two-party inference framework for large transformers, NDSS 2025
> > >
> > > [4] Nimbus: Secure and efficient two-party inference for transformers, Neurips 2024
> > >
> > > [5] Bolt: Privacy-preserving, accurate and efficient inference for transformers, S&P 2024

---

### Official Review · Reviewer_je2K · 2025-03-10

**Overall Recommendation:** 3

**Summary:**

The paper proposes an efficient method for secure inference in generative pre-trained transformer (GPT) models by replacing the traditional autoregressive secure decoding process with a Public decOding and Secure verificaTion (POST) approach. The POST method leverages publicly available GPT models to generate multiple candidate tokens in plaintext, which are then verified securely against a private model. The authors optimize this process through speculative sampling and knowledge distillation, significantly improving token acceptance rates. Their approach achieves between 2.1× and 6.0× speedups compared to traditional secure decoding, without compromising privacy or output quality.

**Claims And Evidence:**

The authors provide convincing evidence supporting their claims of increased inference efficiency and maintained privacy through extensive experimental validation. The speedup claims (2.1× to 6.0×) are backed by thorough performance measurements under different model pairings (Vicuna-7B/LLaMA, FLAN-T5-XL/T5-efficient, FLAN-T5-XL/FLAN-T5-small&base) and network scenarios (LAN/WAN).

**Essential References Not Discussed:**

The references discussed appear comprehensive.

**Experimental Designs Or Analyses:**

The experimental designs are sound and rigorous. The authors provide detailed latency analyses, clearly attributing observed performance improvements to their methodological innovations. The breakdown of latency components (communication, computation, transmission) further reinforces the validity of their findings.

**Methods And Evaluation Criteria:**

The proposed methods and evaluation criteria are appropriate and justified. The authors select a range of representative models (small and large GPT variants) and tasks (text-to-SQL, mathematical reasoning, code generation, finance QA) to demonstrate the broad applicability and effectiveness of their method. The benchmark datasets are standard and widely used, making the evaluation meaningful.

**Other Comments Or Suggestions:**

NA

**Other Strengths And Weaknesses:**

Strengths:
- Clearly articulated motivation and innovative use of public models for secure verification.
- Strong experimental validation demonstrating significant practical improvements.
- Practical protocol optimization for cryptographic operations enhancing real-world applicability.

Weaknesses:
- The approach relies on the acceptance ratio achieved by the public model; if the alignment is suboptimal, the speedup may diminish.
- The complexity of the secure verification protocol may present implementation challenges in real-world deployments.
- Additional exploration of scalability with larger vocabulary sizes or more diverse model architectures would be beneficial.

**Questions For Authors:**

NA

**Relation To Broader Scientific Literature:**

It clearly outlines differences from related approaches, such as BumbleBee, Nimbus, and other cryptographic optimizations, emphasizing the novelty of leveraging public GPT models for secure verification.

**Theoretical Claims:**

The theoretical claims (such as those related to speculative sampling) have been reviewed. The provided protocol for speculative sampling is logically sound, clearly described, and addresses critical performance bottlenecks inherent in secure computation.

---

> ### Author Rebuttal · Authors · 2025-03-30
>
> Thank you for the reviewer's appreciation of our efforts and valuable feedback on our paper. We address the main concerns as follows.
> # Question 1: The approach relies on the acceptance ratio achieved by the public model; if the alignment is suboptimal, the speedup may diminish.
> Our experiments demonstrate a consistent acceptance ratio and speedup across three pairs of models and four tasks. Even under suboptimal conditions, such as utilizing a small, irrelevant public model, the alignment achieves acceptance ratios of 50%–70% (corresponding to a speedup of approximately 2X–3.3X). When employing a small model from the same series, the acceptance ratios improve to 75%–82% (corresponding to a speedup of around 4X–5.5X).
>
> Furthermore, as discussed in Line 293 and Line 370, the acceptance ratio has greater potential than what the experimental results suggest. For instance, our experiments only adopt the 68M or 160M GPT models as the public model. As client-side computational capabilities continue to advance, it is anticipated that the acceptance ratio will increase through the utilization of larger and more powerful public models. Additionally, our experiments only use a very small alignment dataset and do not involve extensive hyper-parameter searching. A more thorough alignment process is expected to yield higher acceptance ratios, particularly if the server provides a more appropriately aligned public model, benefiting from its insights into the training datasets and access to enhanced computational resources.
>
> # Question 2: The complexity of the secure verification protocol may present implementation challenges in real-world deployments.
> The proposed approach reduces the execution time of secure inference by a factor of 2X to 6X, while preserving the same levels of security and accuracy. Although this introduces some additional implementation complexity, we are willing to mitigate this challenge by open-sourcing our implementation.
>
> # Question 3: Additional exploration of scalability with larger vocabulary sizes or more diverse model architectures would be beneficial.
> Thank you for your advice on the scalability analysis. Currently, our experiment includes three pairs of model architectures and four tasks. We focus on these aspects as they are most relevant to the proposed approach. Regarding vocabulary size, we utilize a vocabulary consisting of approximately 30,000 tokens, which provides comprehensive token coverage and is commonly adopted by popular LLMs. The existing experiments are sufficiently convincing to demonstrate the effectiveness of our method, but we are willing to include additional experiments to further justify the scalability concerning vocabulary size and model architectures in our next version.

---

### Official Review · Reviewer_PJwW · 2025-03-13

**Overall Recommendation:** 3

**Summary:**

This paper focus on secure inference on GPT, and presents POST, which contains (1) a private sampling protocol optimized for cryptographic primitives and (2) model alignment using knowledge distillation to speedup the secure inference. Experiments demonstrate speedup compared to standard decoding across three pairs of public-private models and different network conditions.

**Claims And Evidence:**

1. In section 4.2, the author claims that the division is refactored into multiplication (line 267). It is unclear why this can be done. The proof in Appendix D lacks showing the range of p(x)/q(x).

2. In this paper, the author introduces the alignment of public model and private model. If the alignment dataset closely resembles the private dataset (line 303-304), how to evaluate the potential privacy leakage? No evidence is provided towards this.

**Essential References Not Discussed:**

No.

**Experimental Designs Or Analyses:**

1. This paper lacks experimental comparison with prior works [1-2] on secure GPT inference. Despite that the author claims the proposed work is orthogonal to prior works, it is worth to show at least one combination to verify this claim.

[1] Hou, X., Liu, J., Li, J., Li, Y., Lu, W.-j., Hong, C., and Ren, K. Ciphergpt: Secure two-party gpt inference. Cryptology ePrint Archive, 2023.

[2] Gupta, K., Jawalkar, N., Mukherjee, A., Chandran, N., Gupta, D., Panwar, A., and Sharma, R. Sigma: secure gpt inference with function secret sharing. Cryptology ePrint Archive, 2023.

**Methods And Evaluation Criteria:**

The proposed methods and evaluation make sense.

**Other Comments Or Suggestions:**

A minor comment about clarity: Figure 3 is not clear enough for illustrating the proposed methods of this work. There is no information about where the secure inference happens, no input to the private model and public model, not showing 'draft tokens', 'bonus token'. I suggest the authors to enrich Figure 3 and present more details. As the authors claim the proposed POST is an orthogonal approach to prior works (in line 33), it is important to make the framework clearly visualized. Current Figure 3 does not help understand the paper.

**Other Strengths And Weaknesses:**

Strength: The motivation section is well-written and intriguing. Showing latency breakdown in terms of one-way delay, transmission and computation is critical to the community.

**Questions For Authors:**

In this paper, the author introduces the alignment of public model and private model. If the alignment dataset closely resembles the private dataset (line 303-304), how to show/evaluate/quantify whether there is privacy leakage?

**Relation To Broader Scientific Literature:**

The proposed POST is in parallel with prior methods [1-3] in the field of secure inference of GPT, which is a strength of this paper. This paper also discussed recent secure inference on GPT in Appendix B.

[1] Hao, M., Li, H., Chen, H., Xing, P., Xu, G., and Zhang, T. Iron: Private inference on transformers. Advances in Neural Information Processing Systems, 35:15718–15731, 2022.

[2] Zeng, W., Li, M., Xiong, W., Tong, T., Lu, W.-j., Tan, J., Wang, R., and Huang, R. Mpcvit: Searching for accurate and efficient mpc-friendly vision transformer with heterogeneous attention. In Proceedings of the IEEE/CVF International Conference on Computer Vision, pp. 5052–5063, 2023.

[3] Lu, W.-j., Huang, Z., Gu, Z., Li, J., Liu, J., Ren, K., Hong, C., Wei, T., and Chen, W. Bumblebee: Secure two-party inference framework for large transformers. Cryptology ePrint Archive, 2023.

**Theoretical Claims:**

In Appendix D, the authors try to prove the division can be refactored as multiplication. However, it is not stated what is the range of p(x)/q(x). Therefore, I am not sure whether the claim is correctly proved.

---

> ### Author Rebuttal · Authors · 2025-03-30
>
> Thank you for your thorough examination and thoughtful feedback on our paper. We address the main concerns as follows.
> # Question 1: In Appendix D, the authors try to prove the division can be refactored as multiplication. However, it is not stated what is the range of p(x)/q(x). Therefore, I am not sure whether the claim is correctly proved.
> We provide an explanation here and will include more detailed explanations for the derivation in Appendix D to ensure it is easy to follow.
>
> Equation (6) derives the equivalent condition for determining whether a token should be rejected. We assume the reviewer’s concerns is about the elimination of the $min()$ function in the third line $r \geq \min \left(1, \frac{p(\hat{x})}{q(\hat{x})}\right)$. The equivalence of the conditions in line 3 and line 4 is established through two cases based on the range of $\frac{p(\hat{x})}{q(\hat{x})}$.
> - $\frac{p(\hat{x})}{q(\hat{x})} \in [0,1]$: The $min()$ function can be directly removed, as stated in Equation (6).
> - $\frac{p(\hat{x})}{q(\hat{x})} \in (1,\infty)$: The third line simplifies to $r>1$. Since $r$ is drawn from a uniform distribution over the interval $[0,1]$, the conditions in both line 3 and line 4 are never satisfied. Consequently, the equivalence remains valid.
>
> If you have any further questions, please feel free to let us know.
>
>
> # Question 2: If the alignment dataset closely resembles the private dataset (line 303-304), how to evaluate the potential privacy leakage?
> Line 303 discusses the case in which the server provides the aligned public model. We assume the concern is about the open-source aligned public model incurring potential privacy leakage of the alignment dataset. This issue is independent of this work and widely exists. For instance, the base model used for alignment, open-source smaller versions of private models, can similarly expose privacy risks related to the pre-training dataset (private dataset). Such kind of issue can be resolved using the well-established framework known as differential privacy [1]. Differential privacy preserves the utility of the training data while anonymizing it with metrics that quantify the upper bound of privacy leakage. In our alignment scenario, if the alignment data resembles the private dataset (for instance, including private information), the direct use of the original dataset for training is avoided. Instead, a sanitized aligning dataset is generated [2,3], or differential privacy is applied during the gradient descent process [4].
>
> Moreover, the potential for privacy leakage in our alignment is more manageable compared to the potential leakage of pre-training datasets from open-source small models. This is because the alignment's objective is to let the public model mimic the output distribution of easily predicted tokens. This makes the requirements on the alignment dataset much smaller than traditional training: a very small number of data is sufficient (Figure 4), and alignment on simple tokens rarely involves sensitive information.
>
> To avoid any privacy leakage, a cautious server can choose to use relevant publicly available datasets or not provide the alignment. This may result in slightly slower performance, but thanks to our speculative sampling protocol, even non-aligned public models still achieve a 1.5X to 5X speedup.
>
> [1] The Algorithmic Foundations of Differential Privacy, Foundations and Trends in Theoretical Computer Science 2014
> [2] Dp-opt: Make large language model your privacy-preserving prompt engineer, ICLR 2024
> [3] Privacy-Preserving In-Context Learning with Differentially Private Few-Shot Generation, ICLR 2024
> [4] Deep learning with differential privacy, CCS 2016
>
> # Question 3: Lack experimental comparison with prior works (such as CipherGPT and SIGMA) on secure GPT inference. Despite that the author claims the proposed work is orthogonal to prior works, it is worth to show at least one combination to verify this claim.
> Sorry for any confusion regarding our experimental setup. In fact, the speedup in Figure 5 is the comparison with the latest work, both with and without the integration of our method. The baselines are the SOTA 2PC secure inference works for Transformer, as detailed in Line 366, including the linear layer protocol from Nimbus [1] and the non-linear layer protocol from BumbleBee [2]. We will further clarify this in the paper.
> [1] Bumblebee: Secure two-party inference framework for large transformers, NDSS 2025
> [2] Nimbus: Secure and efficient two-party inference for transformers, Neurips 2024
>
> # Question 4: Improve Figure 3 for better clarity.
> Thank you for the valuable suggestion. We will enhance Figure 3 to improve its clarity. For example, we will highlight the workflow of the secure inference process, emphasizing the distinctions between our approach and prior works. We will also illustrate the appearance of the "draft token" and "bonus token" throughout the process to provide a clearer understanding.

---

### Official Review · Reviewer_SCbC · 2025-03-16

**Overall Recommendation:** 3

**Summary:**

To accelerate privacy-preserving inference, the authors propose a Public Decoding and Secure verificaTion (POST) approach that utilizes public GPT models, based on the observation that securely decoding one token vs. multiple tokens takes a similar latency.
Since the efficiency of secure decoding depends on the acceptance rate of tokens proposed by the public model, they purpose two optimizations 1) a private sampling protocol specific to crypto primitives 2) model alignment using knowledge distillation. The optimized approach remains the same privacy level and generation quality while improving up to 6x speed up across different public-private model pairs.

**Claims And Evidence:**

page 2 col 1 line 90-92: “This approach broadly applies across different cryptographic protocols and GPT models, where we observe similar insensitivity.“
In the paper,  the authors clarify how the approach is applied to versatile GPT models, but it’s not clear to me how it can be applied to different cryptographic protocols. Specifically, in the abstract, the authors mention that the speculative sampling protocol is specific to crypto primitives.

**Essential References Not Discussed:**

N/A

**Experimental Designs Or Analyses:**

* The experimental design effectively addresses my initial questions from the 'methods' section, specifically regarding the runtime of the knowledge distillation process and the accuracy improvement post-alignment.
* Could the author elaborate on the choice of the three pairs of public and private models? I understand the inclusion of pairs from different series (the first two pairs) and a pair from the same series (the third pair). However, what is the underlying rationale for selecting two pairs from different series?
* What dataset was used for the knowledge distillation? I am curious whether the acceptance rate was influenced by any similarities between the dataset used for alignment and the dataset used for evaluation.
* Despite the use of different methodologies, I would appreciate a performance comparison with related work.

**Methods And Evaluation Criteria:**

I really enjoy reading the motivation section. The two observations clearly motivate the later-on approaches to optimize the decoding phase in the generation process.

**Other Comments Or Suggestions:**

Typos or Comments:
	page 1 col 2 line 9: missing space after the parenthesis “Pang et al., 2024)lever-“

**Other Strengths And Weaknesses:**

Strengths: The experiment is well-designed and effectively demonstrates the enhancements achieved through the proposed methodologies.

Weaknesses: The contribution of this work may be considered incremental since it primarily utilizes standard techniques such as knowledge distillation and batching multiple tokens.

**Questions For Authors:**

N/A

**Relation To Broader Scientific Literature:**

While other works mainly optimize the protocols or modify the model architectures, this work is complementary to those works and can be integrated for further performance.

**Theoretical Claims:**

* There is the proof for the correctness of the sampling protocol (in the appendix although) and also security analysis for privately reject draft tokens and knowledge distillation.
* page 6 col 1 line 292: it appears that the complexity remains substantial since 2^l represents the size of the field. Could the author clarify why this level of complexity is considered acceptable?

---

> ### Author Rebuttal · Authors · 2025-03-30
>
> We are grateful for the reviewer's appreciation of our efforts. Below, we respond to your constructive comments in detail.
> # Question 1: Could the author clarify the ambiguity in Line 90 in Introduction and the Line 27 in abstract?
> Thank you for pointing out the potential ambiguity. We clarify it here and will revise the text to avoid any further confusion. Line 90 describes the whole POST approach, and line 27 refers to only the proposed sampling protocol.
> - In lines 90-92: We claim POST is compatible with mainstream 2PC protocols for Transformer models (e.g., BOLT [4], BumbleBee [2], Nimbus [3]) because the key observation: the latency insensitivity to input length holds across these protocols, for which we provide more experiments in Appendix F.
> - In line 27: We refer to the efforts of designing speculative sampling protocols in Section 4.2 to eliminate operations that are inefficient for cryptographic primitives.
> # Question 2: Line 292: Could the author clarify why the $2^l$ complexity is considered acceptable?
> The $O(2^l)$ term in the comparison protocol's overhead represents a special case with maximal communication size but minimal communication rounds [1]. Since we focus on optimizing the number of comparison calls rather than the protocol itself, we use this special case as a simplification for easier understanding, and the footnote on page five writes the general form $O(q \cdot 2^m)$ of communication complexity, where $q*m=l$. Here, $O(2^l)$ corresponds to use $q=1$. Larger $q$ reduces communication size but increases rounds. Existing works [2,3] typically choose $q=8$ to balance this trade-off, which we also adopt in experiments. In this way, bit width only partly contributes to exponential complexity.
> # Question 3: What is the underlying rationale for selecting two pairs from different series?
> We design two kinds of experiments to show the effectiveness in different cases.
> - Same-series models: This is the favorable setting in practice, as private models often have open-source smaller versions, which can achieve speedups of 4.2X–6.0X.
> - Different-series models: This is to highlight robustness  and general applicability. Even in less favorable conditions, we still have speedups of 2.1X–4X.
> # Question 4: What dataset was used for the knowledge distillation? Whether the acceptance rate is influenced by any similarities between the alignment dataset and evaluation dataset.
> In this work, the alignment dataset is randomly sampled from the downstream training dataset. As explained in Sec. 4.3, the rationale for using similar datset is the case when client uses relevant public datasets or generates sanitized datasets from his queries (e.g., via differentially private generation [5,6] that preserves utility while removing sensitive information).
>
> Our experiments show that alignment dataset similarity to the downstream task impacts the acceptance rate. For instance, testing Vicuna-7B and LLaMA-160M with an irrelevant alignment dataset, Alpaca [7], to the evaluated tasks.
>
> ||Not-aligned|Irrelevant-aligned|Relevant-aligned|
> |-|-|-|-|
> |SP|0.302|0.372|0.592|
> |GS|0.536|0.602|0.691|
> |CP|0.405|0.463|0.665|
> |FN|0.576|0.595|0.650|
>
> Directly using a non-aligned public models yields a 1.4X–2.3X speedup, while non-relevant dataset slightly improves it to 1.6X–2.5X. The best results (2.5X–2.9X speedup) occur when using a similar alignment dataset. Thus, we recommend selecting an similar alignment dataset for higher acceptance rates.
> # Question 5: Despite the use of different methodologies, I would appreciate a performance comparison with related work.
> Sorry for any confusion regarding our experimental setup. In fact, the speedup in Figure 5 is the comparison with the latest work, both with and without the integration of our method. The baselines are the SOTA 2PC secure inference works for Transformer, as detailed in Line 366, including the linear layer protocol from Nimbus [1] and the non-linear layer protocol from BumbleBee [2]. We will further clarify this in the paper.
> # Question 6: The contribution of this work may be considered incremental since it primarily utilizes standard techniques such as knowledge distillation and batching multiple tokens.
> Our key contributions include being the first to introduce observations on public models and latency insensitivity to input length. The novel paradigm (public decoding and secure verification) is not merely a standard batching technique but a strategically designed approach based on these insights. Additionally, we optimize this paradigm not only through knowledge distillation but also with a specialized sampling protocol.
> # Reference
> [1] Cryptflow2, CCS 2020
>
> [2] Bumblebee, NDSS 2025
>
> [3] Nimbus, Neurips 2024
>
> [4] Bolt, S&P 2024
>
> [5] Dp-opt: Make large language model your privacy-preserving prompt engineer, ICLR 2024
>
> [6] Privacy-Preserving In-Context Learning with Differentially Private Few-Shot Generation, ICLR 2024
>
> [7] https://github.com/tatsu-lab/stanford_alpaca

---

### Decision · Program_Chairs · 2025-05-01

**Decision:**

Accept (poster)

**Comment:**

The paper proposes a public decoding and secure verification method to accelerate privacy-preserving inference. However, some shortcomings need to be addressed in the current version. For example, Figure 3 and its description require a thorough examination, along with a more detailed explanation of the choice of public models. In addition, expanding the experimental evaluation to include larger vocabulary sizes or more diverse model architectures would also improve the paper. After reading the reviewers’ comments and the authors’ rebuttals, I believe these issues can be effectively addressed in the camera-ready version. Given that all reviewers have acknowledged the novelty of the authors' work, I recommend acceptance of the paper.